# ANTGPT: CAN LARGE LANGUAGE MODELS HELP LONG-TERM ACTION ANTICIPATION FROM VIDEOS?

**Qi Zhao**[*]
Brown University

**Shijie Wang**[*]
Brown University

**Ce Zhang**
Brown University

**Changcheng Fu**
Brown University

**Minh Quan Do**
Brown University

**Nakul Agarwal**
Honda Research Institute

**Kwonjoon Lee**
Honda Research Institute

**Chen Sun**
Brown University

## ABSTRACT

Can we better anticipate an actor's future actions (e.g. *mix eggs*) by knowing what commonly happens after the current action (e.g. *crack eggs*)? What if the actor also shares the goal (e.g. *make fried rice*) with us? The long-term action anticipation (LTA) task aims to predict an actor's future behavior from video observations in the form of verb and noun sequences, and it is crucial for human-machine interaction. We propose to formulate the LTA task from two perspectives: a *bottom-up* approach that predicts the next actions autoregressively by modeling temporal dynamics; and a *top-down* approach that infers the goal of the actor and *plans* the needed procedure to accomplish the goal. We hypothesize that large language models (LLMs), which have been pretrained on procedure text data (e.g. recipes, how-tos), have the potential to help LTA from both perspectives. It can help provide the prior knowledge on the possible next actions, and infer the goal given the observed part of a procedure, respectively. We propose **AntGPT**, which represents video observations as sequences of human actions, and uses the action representation for an LLM to infer the goals and model temporal dynamics. **AntGPT** achieves state-of-the-art performance on Ego4D LTA v1 and v2, EPIC-Kitchens-55, as well as EGTEA GAZE+, thanks to LLMs' goal inference and temporal dynamics modeling capabilities. We further demonstrate that these capabilities can be effectively distilled into a compact neural network 1.3% of the original LLM model size. Code and model are available at brown-palm.github.io/AntGPT.

## 1 INTRODUCTION

Our work addresses the long-term action anticipation (LTA) task from video observations. Its desired outputs are sequences of *verb* and *noun* predictions over a typically long-term time horizon for an actor of interest. LTA is a crucial component for human-machine interaction. A machine agent could leverage LTA to assist humans in scenarios such as daily household tasks (Nagarajan et al., 2020) and autonomous driving (Gao et al., 2020). The LTA task is challenging due to noisy perception (e.g. action recognition), and the inherent ambiguity and uncertainty that reside in human behaviors.

A common approach for LTA is *bottom-up*, which directly models the temporal dynamics of human behavior either in terms of the discrete action labels (Sun et al., 2019b), or the latent visual representations (Vondrick et al., 2016). Meanwhile, human behaviors, especially in daily household scenarios, are often "purposive" (Kruglanski & Szumowska, 2020), and knowing an actor's longer-term goal can potentially help action anticipation (Tran et al., 2021). As such, we consider an alternative *top-down* framework: It first explicitly infers the longer-term goal of the human actor, and then *plans* the procedure needed to accomplish the goal. However, the goal information is often left unlabeled and thus latent in existing LTA benchmarks, making it infeasible to directly apply goal-conditioned procedure planning for action anticipation.

Our paper seeks to address these challenges in modeling long-term temporal dynamics of human behaviors. Our research is inspired by prior work on the mental representations of tasks as *action grammars* (Payne & Green, 1986; Pastra & Aloimonos, 2012) in cognitive science, and by large language models' (LLMs) empirical success on procedure planning (Ahn et al., 2022; Driess et al.,

2023). We hypothesize that the LLMs, which use procedure text data during pretraining, encode useful prior knowledge for the long-term action anticipation task. Ideally, the prior knowledge can help both bottom-up and top-down LTA approaches, as they can not only answer questions such as "what are the most likely actions following this current action?", but also "what is the actor trying to achieve, and what are the remaining steps to achieve the goal?"

Concretely, our paper strives to answer four research questions on modeling human behaviors for long-term action anticipation: (1) Does top-down (i.e. goal-conditioned) LTA outperform the bottom-up approach? (2) Can LLMs infer the goals useful for top-down LTA, with minimal additional supervision? (3) Do LLMs capture prior knowledge useful for modeling the temporal dynamics of human actions? If so, what would be a good interface between the videos and an LLM? And (4) Can we condense LLMs' prior knowledge into a compact neural network for efficient inference?

To perform quantitative and qualitative evaluations necessary to answer these questions, we propose **AntGPT**, which constructs an action-based video representation, and leverages an LLM to perform goal inference and model the temporal dynamics. We conduct experiments on multiple LTA benchmarks, including Ego4D (Grauman et al., 2022), EPIC-Kitchens-55 (Damen et al., 2020), and EGTEA GAZE+ (Li et al., 2018). Our evaluations reveal the following observations: First, we find that our video representation, based on sequences of noisy action labels from action recognition algorithms, serves as an effective interface for an LLM to infer longer-term goals, both qualitatively from visualization, and quantitatively as the goals enable a top-down LTA pipeline to outperform its bottom-up counterpart. The goal inference is achieved via in-context learning (Brown et al., 2020), which requires few human-provided examples of action sequence and goal pairs. Second, we observe that the same video representation allows effective temporal dynamics modeling with an LLM, by formulating LTA as (action) sequence completion. Interestingly, we observe that the LLM-based temporal dynamics model appears to perform implicit goal-conditioned LTA, and achieves competitive performance without relying on explicitly inferred goals. These observations enable us to answer the final research question by distilling the bottom-up LLM to a compact student model 1.3% of the original model size, while achieving similar or even better LTA performance.

To summarize, our paper makes the following contributions:

1. We propose to investigate if large language models encode useful prior knowledge on modeling the temporal dynamics of human behaviors, in the context of bottom-up and top-down action anticipation.

2. We propose the **AntGPT** framework, which naturally bridges the LLMs with computer vision algorithms for video understanding, and achieves state-of-the-art long-term action anticipation performance on the Ego4D LTA v1 and v2 benchmarks, EPIC-Kitchens-55, and EGTEA GAZE+.

3. We perform thorough experiments with two LLM variants and demonstrate that LLMs are indeed helpful for both goal inference and temporal dynamics modeling. We further demonstrate that the useful prior knowledge encoded by LLMs can be distilled into a very compact neural network (1.3% of the original LLM model size), which enables efficient inference.

## 2  RELATED WORK

**Action anticipation** can be mainly categorized into next action prediction (NAP) (Damen et al., 2020; Li et al., 2018) and long-term anticipation (LTA) (Grauman et al., 2022). Our work focuses on the LTA task, where modeling the (latent) goals of the actors is intuitively helpful. Most prior works on action anticipation aim at modeling the temporal dynamics directly from visual cues, such as by utilizing hierarchical representations (Lan et al., 2014), modeling the temporal dynamics of discrete action labels (Sun et al., 2019b), predicting future latent representations (Vondrick et al., 2016; Gammulle et al., 2019b), or jointly predicting future labels and features (Girdhar & Grauman, 2021; Girase et al., 2023). As the duration of each action is unknown, some prior work proposed to discover object state changes (Epstein et al., 2021; Souček et al., 2022) as a proxy task for action anticipation. The temporal dynamics of labels or latent representations are modeled by neural networks, and are often jointly trained with the visual observation encoder in an end-to-end fashion. To predict longer sequences into the future for LTA, existing work either build autoregressive generative models (Abu Farha et al., 2018; Gammulle et al., 2019a; Sener et al., 2020; Farha et al., 2020) or use timestep as a conditional parameter and predict in one shot based on provided timestep (Ke et al., 2019). We consider these approaches as bottom-up as they model the shorter-term temporal transitions of human activities.

**Visual procedure planning** is closely related to long-term action anticipation, but assumes that both source state and the goal state are explicitly specified. For example, Chang et al. (2020) proposed to learn both forward and conjugate dynamics models in the latent space, and plans the actions to take accordingly. Procedure planning algorithms can be trained and evaluated with video observations (Chang et al., 2020; Bi et al., 2021; Sun et al., 2022; Zhao et al., 2022; Narasimhan et al., 2023; Bansal et al., 2022), they can also be applied to visual navigation and object manipulation (Driess et al., 2023; Ahn et al., 2022; Lu et al., 2023). Unlike procedure planning, our top-down LTA approach does not assume access to the goal information. Our explicit inference of the high-level goals (with LLMs) also differs from prior attempts to model the goal as a latent variable, which is optimized via weakly-supervised learning (Roy & Fernando, 2022; Mascaro et al.).

**Multimodal learning**, such as joint vision and language modeling, have also been applied to the action anticipation tasks. One approach is to treat the action labels as the language modality, and to distill the text-derived knowledge into vision-based models. For example, Camporese et al. (2021) models label semantics with hand-engineered label prior based on statistics information from the training action labels. Ghosh et al. (2023) trains a teacher model with text input from the training set and distills the text-derived knowledge to a vision-based student model. Sener & Yao (2019) transfers knowledge from a text-to-text encoder-decoder by projecting vision and language representations in a shared space. Compared to these prior work, our focus is on investigating the benefits of large language models for modeling the temporal dynamics of human activities.

## 3 METHOD

We introduce our proposed **AntGPT** framework for LTA. An overview is shown in Figure 1.

### 3.1 LONG-TERM ACTION ANTICIPATION

The long-term action anticipation (LTA) task requires predicting a sequence of $Z$ actions in a long future time horizon based on a video observation. In the LTA task, a long video $V$ is split into an ordered set of $N$ annotated short segments $\{S^j, a^j\}_{j=1}^N$, where $S^j$ denotes the $j$-th segment in video $V$ and $a^j$ denotes the corresponding action label in the form of noun-verb pair $(n^j, v^j)$. The video is also specified with a stop time $T$, which is represented as the index of the last observed segment. In this way, a video is split into the observed segments $V_o$ and the future segments of the video $V_f$ whose labels $\{\hat{a}^{(T+1)}, ..., \hat{a}^{(T+Z)}\}$ are to be predicted. A hyper-parameter $N_{seg}$ controls how many segments the model can observe. Concretely, we take the observable video segments $\{S^j\}_{j=T-N_{seg}+1}^T$ from $V_o$ as input and output action sequence $\{\hat{a}^{(T+1)}, ..., \hat{a}^{(T+Z)}\}$ as predictions. Alternatively, Ego-Topo (Nagarajan et al., 2020) takes a simplified approach, which only requires predicting the set of future actions, but not their ordering.

**Bottom-up and Top-down LTA.** We categorize action anticipation models into bottom-up and top-down. The bottom-up approach directly models the temporal dynamics from the history observations and predicts future actions autoregressively or in parallel. The top-down framework first explicitly infers the longer-term goal from the history actions, then plans the procedure according to both history and the goal. We define the prediction procedure of bottom-up model $\mathcal{F}_{bu}$ as $\{\hat{a}^{(T+1)}, ..., \hat{a}^{(T+Z)}\} = \mathcal{F}_{bu}(V_o)$. Here $a^j$ denotes the $j$-th video segment's action label, and $T$ is the index of the last observed segment. For the top-down model $\mathcal{F}_{td}$, we formulate the prediction procedure into two steps: First, infer the goal by $g = \mathcal{G}_{td}(V_o)$, then perform goal-conditioned planning as $\{\hat{a}^{(T+1)}, ..., \hat{a}^{(T+Z)}\} = \mathcal{F}_{td}(V_o, g)$, where $g$ corresponds to the long-term goal inferred by the top-down model.

### 3.2 VIDEO REPRESENTATION

To understand the benefits of LLMs for video-based LTA, an important design choice is the interface (Zeng et al., 2022; Surís et al., 2023) between visual inputs and the language model. We are interested in investigating how to represent long-form videos in a compact, text-only bottleneck, while being helpful for goal inference and procedure planning with LLMs. The video data often contains complex and dynamic scenarios, with multiple characters, actions, and interactions occurring over an extended period. While such rich information can be potentially captured by (pretrained) visual embeddings or even "video tokens" (Sun et al., 2019a; Wang et al., 2022), it remains unclear what visual representation would be sufficient to compress the long observed video context, while being friendly to the LLMs.

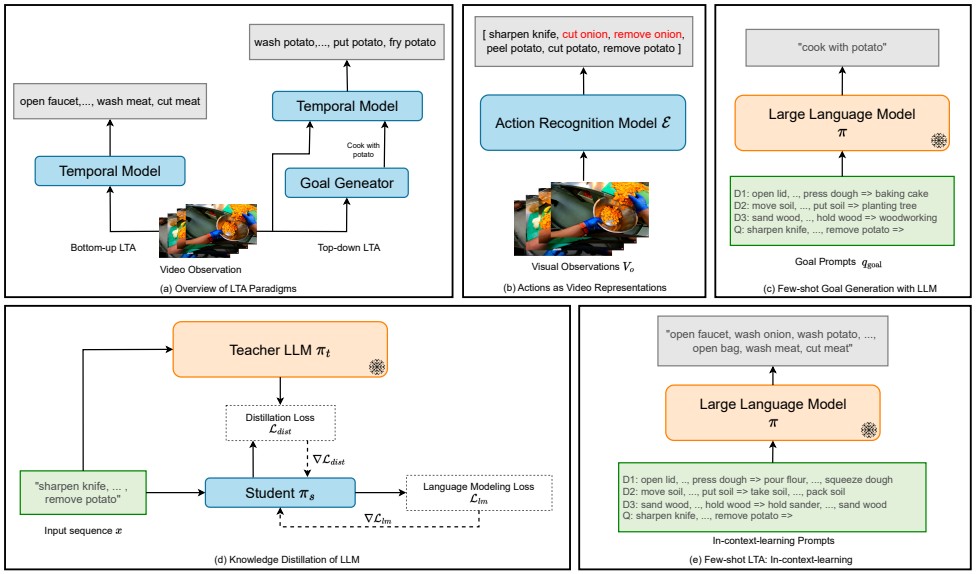

Figure 1: **Illustration of AntGPT.** (a) **Overview of LTA pradigms.** The bottom-up approach predicts future actions directly based on observed human activities, while the top-down approach is guided by high-level goals inferred from observations (hence allows procedure planning). (b) **Actions as video representations.** A pre-trained action recognition model $\mathcal{E}$ takes visual observations $V_o$ as inputs and generates action labels, which can be noisy (shown in red). (c) **Goal inferred by an LLM.** We provide few human-provided examples of action sequences and the expected high-level goals, and leverage an LLM $\pi$ to infer the goal via in-context learning. (d) **Knowledge Distillation.** We distill a frozen LLM $\pi_t$ into a compact student model $\pi_s$ at sequence level. (e) **Few-shot LTA by in-context learning** (ICL), where the ICL prompts can be either bottom-up or top-down.

We first consider the standard approach to represent video frames as distributed embedding representations, computed with pre-trained vision backbone models, such as the CLIP visual encoder (Radford et al., 2021). For each video segment $S^j$ in $V_o$, the backbone extracts the representations of $n$ uniformly sampled frames from this segment to obtain $E^j = \{e_1, e_2, \ldots, e_n\}$. A neural network can then take the embedding representation and predict action labels for the observed frames (action recognition), or the future timesteps (action anticipation).

Our action recognition network $\mathcal{E}$ is implemented as a Transformer encoder. It takes in the visual embeddings and one learnable query token as the input. We then apply two separate MLP heads to decode the verb and noun from the encoded query token. For each observed video segment $S^j$, the recognition model $\mathcal{E}$ takes in randomly sampled image features $E_s^j = \{e_a, e_b, \ldots, e_k\}, E_s^j \subseteq E^j$, and outputs the corresponding action $\hat{a}^{(j)}$ for $S^j$. This process is repeated for every labeled segment in $V_o$, which results in $N_{\text{seg}}$ actions $\{\hat{a}^{(T-N_{\text{seg}})}, \ldots, \hat{a}^{(T)}\}$, in the format of noun-verb pairs. The recognition model $\mathcal{E}$ is trained on the training set to minimize the Cross Entropy Loss between the predictions and the ground-truth action labels.

**How to Represent Videos for the LLMs?** We consider a simple approach to extract video representations for a large language model. We first compute the embedding representation of $V_o$, and then apply the action recognition model $\mathcal{E}$ to convert the distributed representation into discrete action labels, which can be directly consumed by an off-the-shelf LLM. Despite its simplicity, we observe that this representation is strong enough for the LLM to extract meaningful high-level goals for top-down LTA (see Section 3.3), and can even be applied directly to perform both bottom-up and top-down LTA with the LLMs. Alternative approaches, such as discretizing the videos via video captioning or object detection, or projecting the visual embedding via parameter-efficient fine-tuning (Hu et al., 2021; Merullo et al., 2023), can also be applied under our framework.

### 3.3 AntGPT: Long-term Action Anticipation with LLMs

We now describe **AntGPT** (Action **Ant**icipation **GPT**), a framework that incorporates LLMs for the LTA task. An LLM serves both as an few-shot high-level goal predictor via in-context learning, and also as a temporal dynamics model which predicts the future actions conditioned on the observed actions. It hence benefits top-down and bottom-up LTA, in full-shot and few-shot scenarios.

**Few-shot Goal Inference.** In order to perform top-down long-term action anticipation, we conduct in-context learning on LLMs to infer the goals by taking the recognized action labels as inputs, as illustrated in Figure 1 (b) and (c). The ICL prompts $q_{\text{goal}}$ is formulated with examples in the format of `"<observed actions> => <goal>"` and the final query in the format of `"<observed actions> =>"`. The observed actions for the in-context examples are based on ground-truth annotations, and the observed actions in the final query are generated by recognition models. Since no ground truth goals are available, we either use the video metadata as *pseudo goals* when it is available, or design the goals manually. Figure 2 shows several examples for in-context goal inference with the LLM. We treat the raw output of the LLM $T_{\text{goal}} = \pi(q_{\text{goal}})$ as the high-level goal.

**Bottom-up and Top-down LTA.** We now describe a unified framework to perform bottom-up and top-down LTA. The framework largely resembles the action recognition network $\mathcal{E}$ which takes visual embeddings as inputs, but has a few important distinctions. Let's first consider the bottom-up model $\mathcal{B}$. Its transformer encoder takes sub-sampled visual embeddings $E_s^j$ from each segment $S^j$ of $V_o$. The embeddings from different segments are concatenated together along the time axis to form the input tokens to the transformer encoder. To perform action anticipation, we append additional learnable query tokens to the input sequence of the Transformer encoder, each of which corresponds to a future step to predict. Each encoded query token is decoded into verb and noun predictions with two separate MLP heads. We minimize the Cross Entropy Loss for all future actions to be predicted with equal weights. Note that one can choose to use either bidirectional or causal attention masks for the query tokens, resulting in parallel or autoregressive action prediction. We observe that this design choice has marginal impact on performance, and use parallel decoding unless otherwise mentioned.

Thanks to few-shot goal inference with in-context learning, implementing the top-down model $\mathcal{F}_{\text{td}}$ is straightforward: We first embed the inferred goals $T_{\text{goal}}$ with a pre-trained CLIP text encoder. The goal token is then prepended at the beginning of the visual embedding tokens to perform goal-conditioned action anticipation. During training, we use ground-truth action labels to infer the goals via in-context learning. During evaluation, we use the recognized action labels to infer the goals.

**Modeling Temporal Dynamics with LLMs.** We further investigate if LLMs are able to model temporal dynamics via recognized action labels and perform action anticipation via autoregressive sequence completion. We first study the fully supervised scenario, where we perform parameter-efficient (optionally) fine-tuning on LLMs on the training set of an LTA benchmark. Both the input prompt and the target sequence are constructed by concatenating the action labels separated with commas. During training, the input sequences are formed either via teacher forcing (ground truth actions), or the (noisy) recognized actions. The LLM is optimized with the standard sequence completion objective. During inference, we use the action recognition model $\mathcal{E}$ to form input prompts from the recognized action labels. We perform postprocessing to convert the output sequence into action labels. Details of the postprocessing can be found in Section C.1. To perform top-down LTA, we simply prepend an inferred goal at the beginning of each input prompt. The goals are again inferred from ground-truth actions during training, and recognized actions during evaluation.

**Knowledge Distillation** (Hinton et al., 2015) is applied to understand if the knowledge encoded by LLMs about temporal dynamics can be condensed into a much more compact neural network for efficient inference. For sequence models such as LLMs, the distillation loss is calculated as the sum of per-token losses between the encoded feature (e.g. logits) sequences by the teacher and the student. Formally, during distillation, given the input sequence $x$ of length $N$, a well-trained LLM as the teacher model $\pi_t$, the student model $\pi_s$ is optimized to minimize the language modeling loss $\mathcal{L}_{\text{lm}}$ and distillation loss $\mathcal{L}_{\text{dist}} = \sum_{i=1}^{N} D_{KL}(\hat{y}_t^{(i)} || \hat{y}_s^{(i)})$, where $\hat{y}_t = \pi_t(x)$ and $\hat{y}_s = \pi_s(x)$ are the feature sequence encoded by $\pi_t$ and $\pi_s$ respectively, $i$ is the token index of the target sequence, and $D_{KL}$ is the Kullback-Leibler divergence between the teacher and student distribution. The teacher model $\pi_t$ is frozen during training. An illustration is shown in Figure 1 (d).

**Few-shot Learning with LLMs.** Beyond fine-tuning, we are also interested in understanding if LLM's in-context learning capability generalizes to the LTA task. Compared with fine-tuning model with the whole training set, in-context learning avoids updating the weights of a pre-trained LLM. As illustrated in Figure 1 (e), an ICL prompt consists of three parts: First, an instruction that specifies the anticipating action task, the output format, and the verb and noun vocabulary. Second, the in-context examples randomly sampled from the training set. They are in the format of `"<observed actions> => <future actions>"` with ground-truth actions. Finally, the query in the format `"<observed actions> => "` with recognized actions. An example of

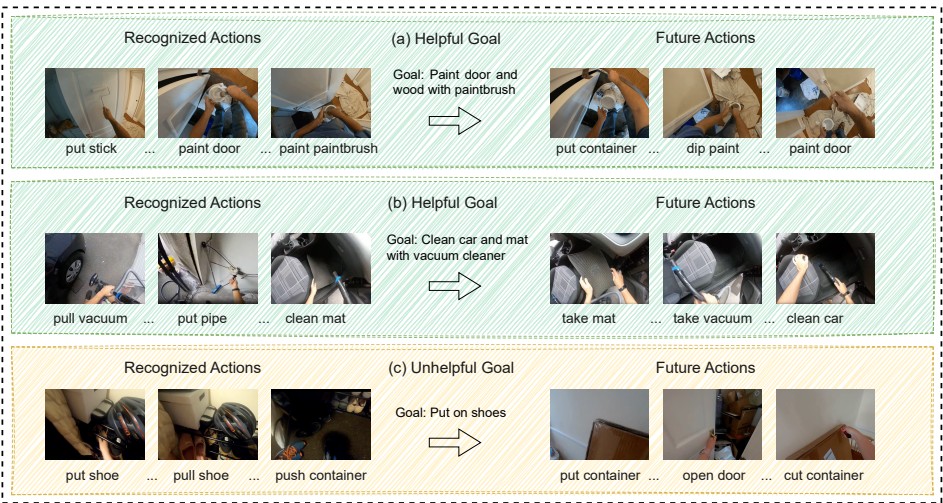

Figure 2: **Examples of the goals inferred by LLMs.** Goals are inferred from the recognized actions of the 8 observed segments. The future actions are ground truth for illustration purposes.

the model's input and output is shown in Figure A1 (b). Alternatively, we also attempt to leverage chain-of-thoughts prompts (Wei et al., 2022) (CoT) to ask the LLM first infer the goal, then perform LTA conditioned on the inferred goal. An example of CoT LTA is shown in Figure A1 (c).

## 4 EXPERIMENTS

We now present quantitative results and qualitative analysis on the Ego4D (Grauman et al., 2022), EPIC-Kitchens (Damen et al., 2020), and EGTEA Gaze+ (Li et al., 2018) benchmarks.

### 4.1 EXPERIMENTAL SETUP

**Ego4D** Grauman et al. (2022) The *v1* dataset contains 3,670 hours of egocentric video of daily life activity spanning hundreds of scenarios. We focus on the videos in the *Forecasting* subset which contains 1723 clips with 53 scenarios. The total duration is around 116 hours. There are 115 verbs and 478 nouns in total. The *v2* dataset contains 3472 annotated clips with total duration of around 243 hours. There are 117 verbs and 521 nouns. We follow the datasets' standard splits.

**EPIC-Kitchens-55** (Damen et al., 2020) (EK-55) contains 55 hours egocentric videos of cooking activities of different video takers. Each video is densely annotated with action labels, spanning over 125 verbs and 352 nouns. We adopt the train and test splits from Nagarajan et al. (2020).

**EGTEA Gaze+** (Li et al., 2018) (EGTEA) contains 86 densely labeled egocentric cooking videos over 26 hours. There are 19 verbs and 53 nouns. We adopt the splits from Nagarajan et al. (2020).

**Evaluation Metrics.** For Ego4D, we use the edit distance (ED) metric. It is computed as the Damerau-Levenshtein distance over sequences of predictions of verbs, nouns or actions. We follow the standard practice in Grauman et al. (2022) and report the minimum edit distance between each of the top $K = 5$ predicted sequences and the ground-truth. We report Edit Distance at $Z = 20$ (ED@20) on the validation set and the test set. For EK-55 and EGTEA, we follow the evaluation metric described in Nagarajan et al. (2020). The first K% of each video is given as input, and the goal is to predict the set of actions happening in the remaining (100-K)% of the video as multi-class classification. We sweep values of K = [25%, 50%, 75%] representing different anticipation horizons and report mean average precision (mAP) on the validation sets. We report the performances on all target actions (All), the frequently appeared actions (Freq), and the rarely appeared actions (Rare) as in Nagarajan et al. (2020). A number of previous work reported performance on these two datasets. The order agnostic LTA setup in these two datasets complements the Ego4D evaluation.

**Implementation Details.** We use the frozen CLIP (Radford et al., 2021) ViT-L/14 for image features, and a transformer encoder with 8 attention heads, and 2048 hidden size for the recognition model. To study the impact of vision backbones, we also include EgoVLP, a video backbone pre-trained on Ego4D datasets. For the large language models, we adopt open-source Llama2-13B for in-context learning and 7B model for fine-tuning. For comparison, we also use OpenAI's GPT-3.5 Turbo for in-context learning and GPT-3 curie for fine-tuning. More details and ablation study on recognition model, teacher forcing, LLMs and other design choices are described in appendix.

## 4.2 CAN LLMs INFER GOALS TO ASSIST TOP-DOWN LTA?

We compare two LLMs, GPT-3.5 Turbo and Llama2-chat-13B, on goal inference: To obtain the pseudo ground-truth goals for constructing the in-context examples, we use the video titles for EGTEA, and the video descriptions for EK-55. We manually annotate the goals for Ego4D. We use 12 in-context examples to infer the goals. For EK-55 and EGTEA, we always use the recognized actions in the first 25% of each video to infer the goals. For Ego4D, we set $N_{\text{seg}} = 8$.

We first use the Transformer encoder model described in Section 3.3 as the temporal model: It allows us to study the standalone impact of goal conditioning by comparing the bottom-up and the top-down LTA performances. The Transformer encoder takes in the same visual features as used for action recognition. The text embeddings of the inferred goals are provided for the top-down variant. Table 1 shows results on Ego4D v1, EK-55, and EGTEA. We notice a clear trend that using the inferred goals leads to consistent improvements for the top-down approach, especially for the *rare* actions of EK-55 and EGTEA. We also noticed that both LLMs are able to infer helpful goals for top-down LTA and GPT-3.5 Turbo generates goals slightly better than the ones from Llama2-chat-13B. We also construct "oracle goals" using the video metadata provided by EK-55 and EGTEA datasets. We observe that using the oracle goals leads to slight improvements, indicating that the inferred goals already offer competitive performance improvements. Figure 2 provides some examples of the helpful and unhelpful goals inferred by Llama2.

| Method | Ego4d v1 (ED) | | EK-55 (mAP) | | | EGTEA (mAP) | | |
|---|---|---|---|---|---|---|---|---|
| | Verb ↓ | Noun ↓ | ALL ↑ | Freq ↑ | Rare ↑ | ALL ↑ | Freq ↑ | Rare ↑ |
| image features | 0.735 | 0.753 | 38.2 | **59.3** | 29.0 | 78.7 | 84.7 | 68.3 |
| image features + Llama2 inferred goals | 0.728 | 0.747 | **40.1** | 58.1 | **32.1** | 80.0 | 84.6 | 70.0 |
| image features + GPT-3.5 inferred goals | **0.724** | **0.744** | **40.1** | 58.8 | 31.9 | **80.2** | **84.8** | **72.9** |
| image features + oracle goals* | - | - | 40.9 | 58.7 | 32.9 | 81.6 | 86.8 | 69.3 |

Table 1: **Impact of goal conditioning on LTA performance.** Goal-conditioned (top-down) models outperforms the bottom-up model in all three datasets. We report edit distance for Ego4D, mAP for EK-55 and EGTEA. All results are reported on the validation set.

## 4.3 DO LLMs MODEL TEMPORAL DYNAMICS?

| Model | Goal | Input | Verb ↓ | Noun ↓ |
|---|---|---|---|---|
| Transformer | GPT-3.5 | image features | 0.724 | 0.744 |
| GPT-3-curie | GPT-3.5 | recog actions | 0.709 | 0.729 |
| Transformer | Llama2-13B | image features | 0.728 | 0.747 |
| Llama2-7B | Llama2-13B | recog actions | **0.700** | **0.717** |

Table 2: **Comparison of temporal models for top-down LTA.** Results on Ego4D v1 val set.

| Model | Goal | Verb ↓ | Noun ↓ |
|---|---|---|---|
| GPT-3-curie | No | **0.707** | **0.719** |
| GPT-3-curie | Yes | 0.709 | 0.729 |
| Llama2-7B | No | 0.704 | **0.705** |
| Llama2-7B | Yes | **0.700** | 0.717 |

Table 3: **Top-down vs Bottom-up for LLM-based LTA.** Results on v1 val set.

We further explore if LLMs can be directly applied to model temporal dynamics. We focus on the Ego4D benchmark as it measures the ordering of the anticipated actions.

**LLMs are able to model temporal dynamics.** To utilize an LLM to predict future actions, we adopt the same video representation as used for in-context goal inference but fine-tune the LLM on the training set. For bottom-up LTA, we by default perform teacher forcing during training, and concatenate the $N_{\text{seg}}$ ground-truth action labels as the input sequence. $Z$ ground-truth action labels are concatenated as the target sequence. During evaluation, we concatenate $N_{\text{seg}}$ recognized actions as input, and postprocess the output sequence into $Z$ anticipated actions. For top-down LTA, we prepend the inferred goals to the input sequence.

We conduct top-down LTA with the open-sourced Llama2-7B LLM. During training, we adopt parameter-efficient fine-tuning (PEFT) with LoRA (Hu et al., 2021) and 8-bit quantization. We compare with the transformer baseline with image features, and report results on Ego4D v1 validation set in Table 2. We observe that leveraging the LLM as the temporal dynamics model leads to significant improvement, especially for nouns. Additionally, we validate that simply adding more layers (and hence increasing the model size) does not improve the performance of the image feature baseline (see Table A1 in appendix), confirming that the improvement comes from the action

representation and better temporal dynamics modeling. The results demonstrate the effectiveness of action-based representation, when an LLM is used for temporal dynamics modeling.

**LLMs can perform few-shot temporal modeling.** We further tested LLMs' ability to model temporal dynamics when only shown a few examples. We consider both in-context learning (ICL) and chain-of-thoughts (CoT) and compare them with a transformer model trained from-scratch with the same examples. The results are illustrated in Table A6. We observed that LLMs can model temporal dynamics competitively in a few-shot setting. As expected, chain-of-thoughts outperforms regular in-context learning, but both significantly outperform fine-tuning the Transformer model.

**LLM-based temporal model performs *implicit* goal inference.** We have shown that LLMs can assist LTA by providing the inferred goals, and serving as the temporal dynamics model, respectively. Does combining the two lead to further improved performance? Table 3 aims to answer this question. We report results with fine-tuned Llama2-7B and GPT-3-curie as the temporal model, which use Llama2-Chat-13B and GPT-3.5 Turbo for goal inference, respectively. We empirically observe that the bigger models lead to better inferred goals, while the smaller models are sufficient for temporal modeling. We observe that the bottom-up performance without explicitly inferred goals are on par (marginally better) with the top-down models for both LLMs. This indicates the LLM may implicitly inferred the goals when asked to predict the future actions, and performing explicit goal inference is not necessary. In the following experiments, we stick with this *implicit* goal inference setup.

| Seq Type | Verb ↓ | Noun ↓ | Action ↓ | Model | Setting | Verb ↓ | Noun ↓ | Action ↓ |
|---|---|---|---|---|---|---|---|---|
| Action Labels | **0.6794** | **0.6757** | **0.8912** | 7B | Pre-trained | 0.6794 | 0.6757 | 0.8912 |
| Shuffled Labels | 0.6993 | 0.6972 | 0.9040 | 91M | From-scratch | 0.7176 | 0.7191 | 0.9117 |
| Label Indices | 0.7249 | 0.6805 | 0.9070 | 91M | Distilled | **0.6649** | **0.6752** | **0.8826** |

Table 4: **Benefit of language prior.** Results on Ego4D v2 test set. We replace original action sequences to semantically nonsensical sequences.

Table 5: **LLM as temporal model.** Results on Ego4D v2 test set. Llama2-7B model is fine-tuned on Ego4D v2 training set. 91M models are randomly initialized.

**Language prior encoded by LLMs benefit LTA.** We further investigate if the *language* (e.g. goals and action labels) used for our video representation is actually helpful to utilize the language priors encoded by the LLMs. We first conduct experiments by replacing the action label representation with two representations that we assume the pretrained LLMs are unfamiliar with: (1) **Shuffled Labels.** We randomly generate a mapping of verbs and nouns so that the original verbs/nouns are 1-to-1 projected to randomly sampled words in the dictionary to construct semantically nonsensical language sequence (e.g "open window" to "eat monitor"). (2) **Label Indices.** Instead of using words to represent actions in the format of verb-noun pairs, we can also use the index of the verb/noun in the dictionary to map the words to digits to form the input and output action sequence.

We fine-tune the Llama2-7B model on the three types of action representations on the Ego4D v2 dataset and report results on the test set. As shown in Table 4, the performance drops severely when shuffled action labels or label indices are used, especially for verb. The performance gap indicates that even LLMs have strong capability to model patterns beyond natural language (Mirchandani et al., 2023), the encoded language prior from large-scale pre-training still significantly benefits long-term video action anticipation.

**LLM-encoded knowledge can be condensed into a compact model.** We first introduce the baseline model Llama2-91M, which is a 6-layer randomly initialized transformer decoder model with the similar structure as Llama2-7B. The 91M model takes in the same input during training and evaluation and follows the same post-processing. We then conduct model distillation to use the Llama2-7B model tuned on Ego4D v2 training set as the teacher model and the same randomly initialized Llama2-91M as the student model. Results on test set are shown in Table 5. We observe that the distilled model achieves significant improvement comparing with model trained without distillation in the second row (7.3% and 6.1% for verb and noun). It's also worth noting that the distilled 91M model even outperforms the 7B teacher model on all three metrics, while using 1.3% of the model size. The results confirm that LLM-encoded knowledge on *implicit* goal inference and *explicit* temporal modeling can be condensed into a compact neural network.

| Method | Version | Verb ↓ | Noun ↓ | Action ↓ |
|---|---|---|---|---|
| HierVL (Ashutosh et al., 2023) | v1 | 0.7239 | 0.7350 | 0.9276 |
| ICVAE(Mascaro et al.) | v1 | 0.7410 | 0.7396 | 0.9304 |
| VCLIP (Das & Ryoo, 2022) | v1 | 0.7389 | 0.7688 | 0.9412 |
| Slowfast (Grauman et al., 2022) | v1 | 0.7389 | 0.7800 | 0.9432 |
| **AntGPT** (ours) | v1 | **0.6584**±7.9e-3 | **0.6546**±3.8e-3 | **0.8814**±3.1e-3 |
| Slowfast (Grauman et al., 2022) | v2 | 0.7169 | 0.7359 | 0.9253 |
| VideoLLM (Chen et al., 2023) | v2 | 0.721 | 0.725 | 0.921 |
| PaMsEgoAI (Ishibashi et al., 2023) | v2 | 0.6838 | 0.6785 | 0.8933 |
| Palm (Huang et al., 2023) | v2 | 0.6956 | 0.6506 | 0.8856 |
| **AntGPT** (ours) | v2 | **0.6503**±3.6e-3 | **0.6498**±3.4e-3 | **0.8770**±1.2e-3 |

Table 6: **Comparison with SOTA methods on the Ego4D v1 and v2 test sets in ED@20.** Ego4d v1 and v2 share the same test set. V2 contains more training and validation examples than v1.

| Method | EK-55 | | | EGTEA | | |
|---|---|---|---|---|---|---|
| | ALL | FREQ | RARE | ALL | FREQ | RARE |
| I3D (Carreira & Zisserman, 2017) | 32.7 | 53.3 | 23.0 | 72.1 | 79.3 | 53.3 |
| ActionVLAD (Girdhar et al., 2017) | 29.8 | 53.5 | 18.6 | 73.3 | 79.0 | 58.6 |
| Timeception (Hussein et al., 2019a) | 35.6 | 55.9 | 26.1 | 74.1 | 79.7 | 59.7 |
| VideoGraph (Hussein et al., 2019b) | 22.5 | 49.4 | 14.0 | 67.7 | 77.1 | 47.2 |
| EGO-TOPO (Nagarajan et al., 2020) | 38.0 | 56.9 | 29.2 | 73.5 | 80.7 | 54.7 |
| Anticipatr (Nawhal et al., 2022) | 39.1 | 58.1 | 29.1 | 76.8 | 83.3 | 55.1 |
| **AntGPT** (ours) | **40.1**±2e-2 | **58.8**±2e-1 | **31.9**±5e-2 | **80.2**±2e-1 | **84.8**±2e-1 | **72.9**±1.2 |

Table 7: **Comparison with SOTA methods on the EK-55 and EGTEA Dataset in mAP.** ALL, FREQ and RARE represent the performances on all, frequent, and rare target actions respectively.

## 4.4 COMPARISON WITH STATE-OF-THE-ART

Finally, we compare AntGPT with the previous state-of-the-art methods. We choose the model design settings such as recognition models and input segments number based on ablation study discussed in appendix Section A. For Ego4d v1 and v2, we train the action recognition and fine-tune the LLM temporal models with their corresponding training set. Table 6 shows performance comparisons on Ego4D v1 and v2 benchmarks. We observe that AntGPT achieves best performance on both datasets and largely outperforms other SoTA baselines. Since Ego4d v1 and v2 share the same test set, it is also worth mentioning that our model trained solely on v1 data is able to outperform any other models trained on the v2 data, which indicates the data efficiency and the promise of our approach.

For EK-55 and EGTEA, we compare the goal-conditioned AntGPT with the previous state-of-the-art results in Table 7. AntGPT achieves the overall best performance on both datasets. We observe that our proposed model performs particularly well on rare actions.

## 5 CONCLUSION AND FUTURE WORK

In this paper, we propose **AntGPT** to investigate if large language models encode useful prior knowledge on bottom-up and top-down long-term action anticipation. Thorough experiments with two LLM variants demonstrate that LLMs are capable of inferring goals helpful for top-down LTA and also modeling the temporal dynamics of actions. Moreover, the useful encoded prior knowledge from LLMs can be distilled into very compact neural networks for efficient practical use. Our proposed method sets new state-of-the-art performances on the Ego4D LTA, EPIC-Kitchens-55, and EGTEA GAZE+ benchmarks. We further study the advantages and limitations of applying LLM on video-based action anticipation, thereby laying the groundwork for future research in this field.

**Limitations.** Although our approach provides a promising new perspective in tackling the LTA task, there are limitations that are worth pointing out. The choice of representing videos with fixed-length actions is both efficient and effective for LTA task. However, the lack of visual details may pose constraints on other tasks. Another limitation is the prompt designs of ICL and CoT are still empirical, and varying the prompt strategy may cause significant performance differences. Finally, as studied in our counterfactual experiments, the goal accuracy would have significant impact on the action recognition outputs, and an important future direction is to improve the inferred goal accuracy, and also take multiple plausible goals into account.

**Acknowledgements.** We would like to thank Nate Gillman for valuable feedback. This work is in part supported by Honda Research Institute, Meta AI, and Samsung Advanced Institute of Technology.

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

# A    ABLATION STUDY

In this section, we conduct ablation study on model designs for the Transformer model and LLM-based model on Ego4D v1 and v2 to show how we choose the final settings. In the table, rows labeled as gray denote the final setting we use to report in Table 6.

## A.1    TRANSFORMERS: NUMBER OF LAYERS

We want to compare fine-tuned LLM with auto-regressive transformers trained from-scratch. However, our transformer model and the large language model we used do not have comparable parameter size. Here, we want to explore how parameter size affect the performance of the transformer and to investigate whether the performance gap we demonstrated in 4.4 is due to model size. We ran the transformer model with varying number of layers from 1 to 6 and the corresponding results are in Table A1. We observed that adding more layers to the auto-regressive transformer model does not help performance. This further substantiate our conclusion that large language models are better in modeling action temporal dynamics.

| # layers | Verb $\downarrow$ | Noun $\downarrow$ |
|:---:|:---:|:---:|
| 1 | **0.739** | 0.758 |
| 2 | 0.741 | 0.756 |
| 3 | 0.743 | 0.756 |
| 4 | 0.742 | **0.755** |
| 5 | 0.743 | 0.757 |
| 6 | 0.747 | 0.759 |

Table A1: **Ablations on the number of layers to auto-regressive transformers.** Transformers are trained with actions labels from-scratch. Results on Ego4d v1 validation set. Layer number of Transformer model barely influence its performance.

## A.2    TEACHER FORCING

We developed a data augmentation technique to further boost the performance of our approach. When reporting on test set, we can augment the training set with recognized action tokens which comes from a recognition model that is trained on the validation set and makes inference on the training set. In this way, we double our training set with more noisy data, the distribution of which is closer to the recognized action tokens that we input into the fine-tuned model when making inference on the test set. TableA4 shows the comparison between three training paradigms.

## A.3    RECOGNITION MODEL

To study the impact of vision backbones, we altered the vision features from CLIP to EgoVLP and trained on the same architecture. The result is shown in TableA2 We observed that recognition models based EgoVLP features outperformed CLIP features. This is expected since EgoVLP is pretrained on Ego4D datasets. This finding suggests that our approach is robust to different recognition models and can be further improved with better recognition models.

## A.4    LARGE LANGUAGE MODELS

We compared different LLMs for their performance on fine-tuning. TableA3 shows the result on Ego4D v2 test set. We observe that Llama2-7B with PEFT performs better than fine-tuned GPT-3.

## A.5    NUMBER OF INPUT SEGMENTS

Table A5 shows the influence of the number of input segments to LLM. A higher number of input segments allows the LLM to observe longer temporal context. We choose the input segments from $\{1, 3, 5, 8\}$. We did not go beyond 8 because that was the maximum segments allowed by the Ego4d benchmark. In general, **AntGPT**'s performances on both verb and noun increase as we increase $N_{\text{seg}}$. This is different from the vision baseline in Table 1, whose performance saturates at $N_{\text{seg}} = 3$.

| Recog Model | Verb ↓ | Noun ↓ | Action ↓ |
|---|---|---|---|
| CLIP | 0.6794 | 0.6757 | 0.8912 |
| EgoVLP | **0.6677** | **0.6607** | **0.8854** |

Table A2: **Ablation on recognition model.** Results on Ego4D v2 test set. EgoVLP as recognition model brings better performance.

| Temporal Model | Verb ↓ | Noun ↓ | Action ↓ |
|---|---|---|---|
| GPT-3 curie | 0.6969 | 0.6813 | 0.9060 |
| Llama2-7B | **0.6794** | **0.6757** | **0.8912** |

Table A3: **Ablation on LLM as temporal model.** Results on Ego4D v2 test set. Llama2-7B perform better than GPT-3 curie as tempral model.

| Input | Verb ↓ | Noun ↓ | Action ↓ |
|---|---|---|---|
| gt | 0.6794 | 0.6757 | 0.8912 |
| recog | 0.6618 | 0.6721 | 0.8839 |
| gt+recog | **0.6611** | **0.6506** | **0.8778** |

Table A4: **Ablation on teacher forcing.** Results on Ego4D v2 test set. Using ground-truth and recognized actions as training samples bring best performance.

| # seg | Verb ↓ | Noun ↓ |
|---|---|---|
| 1 | 0.734 | 0.748 |
| 3 | 0.717 | 0.726 |
| 5 | 0.723 | 0.722 |
| 8 | **0.707** | **0.719** |

Table A5: **Ablations on input segments number to LLM.** Results on Ego4d v1 validation set. Segment number of 8 brings best performance.

# B    FEW-SHOT ACTION ANTICIPATION

| Model | Learning Method | With Goal | Verb ↓ | Noun ↓ |
|---|---|---|---|---|
| Transformer | SGD | No | 0.770 | 0.968 |
| Llama2-chat-13B | ICL | No | 0.761 | 0.803 |
| GPT-3.5 | ICL | No | 0.758 | 0.728 |
| GPT-3.5 | ICL | Yes | 0.775 | 0.728 |
| GPT-3.5 | CoT | Yes | **0.756** | **0.725** |

Table A6: **Few-shot results with LLM on Ego4D v1 validation set.** The transformer baseline is trained with the same 12 examples from training set as the in-context examples for LLM.

We conduct quantitative few-shot learning experiments on the Ego4D v1 validation set, and compare the transformer-based model and LLM. The transformer baseline is trained on the same 12 examples sampled from the training set of Ego4D as the in-context examples used by LLM. The transformer is optimized by gradient descent. For top-down LTA with LLM, we compare the two-stage approach to infer goals and predict actions separately, and the one-stage approach based on chain-of-thoughts. Results are shown in Table A6. We observe that all LLM-based methods perform much better than the transformer baseline for few-shot LTA. Among all the LLM-based methods, top-down prediction with CoT prompts achieves the best performance on both verb and noun. However, the gain of explicitly using goal conditioning is marginal, similar to what we have observed when training on the full set. In Figure A1 (b) and (c), we illustrate the example input and output for ICL and CoT, respectively. More examples can be found in Section D.

## B.1    COUNTERFACTUAL PREDICTION

To better understand the impact of goals for action anticipation, we design a "counterfactual" prediction experiment: We first use GPT-3.5-Turbo to infer goals for examples as we did above and treat the outputs as the originally inferred goals, and then manually provide "altered goals" which are different than the original goals but are equally reasonable. We then perform in-context learning using both sets of the goals and with the same sequences of recognized actions as inputs to the LLM. Figure A2 illustrates our workflow and two examples. In the second example, we can see that although the recognized actions are the same, the output action predictions are all different. Despite the differences, using "fix machine" as the goal generates actions highly related to fixing such as "tighten nut" and "turn screw" while switching to "examine machine" leads to actions like "test machine" and "record data". More examples can be found in Section D.

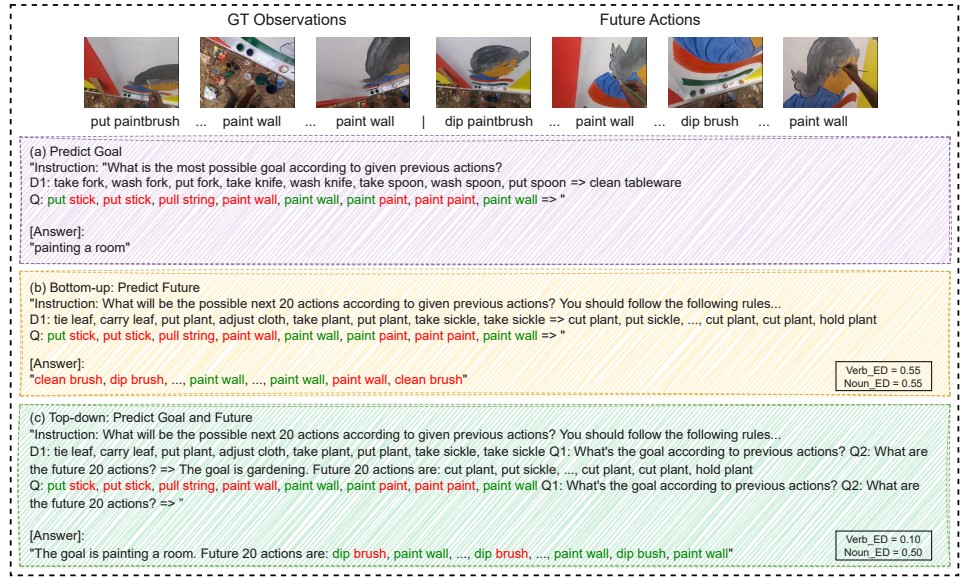

Figure A1: **Illustration of goal prediction and LTA with LLMs**: (a) High-level goal prediction wth in-context learning (ICL). (b) Few-shot bottom-up action prediction with ICL. (c) Top-down prediction with chain-of-thoughts (CoT). The green word indicates correctly recognized actions (inputs to the LLM) and future predictions (outputs of the LLM), red indicates incorrectly recognized or predicted actions. For this example, the ground-truth observations are `[put paintbrush, adjust paintbrush, take container, dip container, paint wall, paint wall, dip wall, paint wall]`.

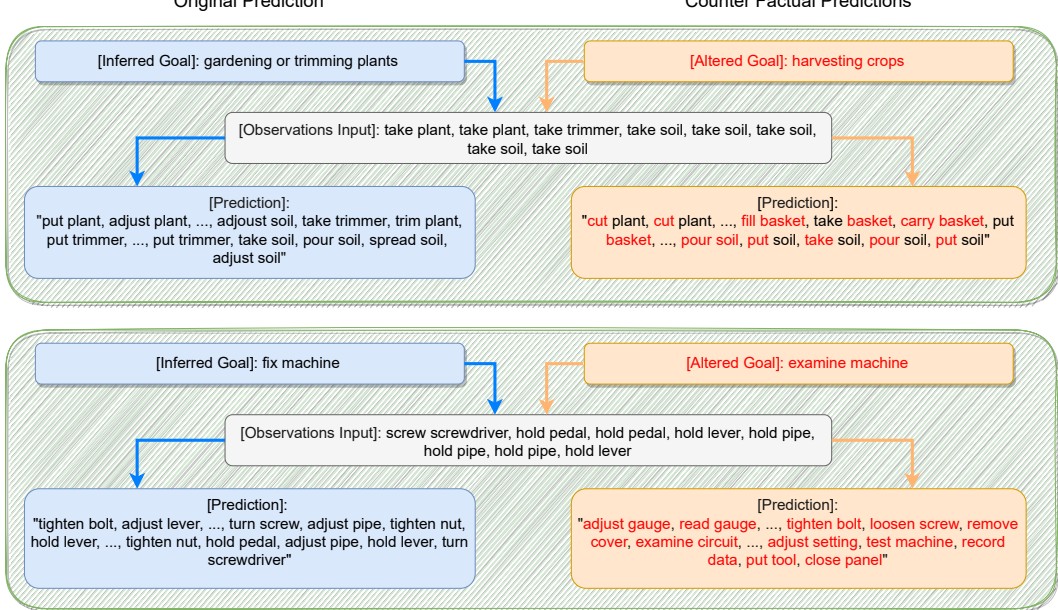

Figure A2: **Illustrations of *counterfactual* prediction.** We replace the originally inferred goal (*gardening or trimming plants* and *fix machine*) with an altered goal (*harvesting crops* and *examine machine*), and observe that the anticipated actions change accordingly, even with the same set of recognized actions as the inputs to the LLM. Words marked in red highlight the different predictions.

Our qualitative analysis with the counterfactual prediction shows that the choice of goal can have large impacts on the action anticipation outputs. This indicates another future direction to improve

the top-down action anticipation performance is to improve the accurate and quality of the predicted goals, and to consider multiple plausible goals.

## C ADDITIONAL EXPERIMENTAL DETAILS

In this section, we provide additional details for the fine-tuning language model experiments on Ego4D v1 and v2, as well as goal inference via in-context learning. We also provide additional details for all the transformer models involved in experiments on all the datasets. Finally, we describe the details of the different setups in Ego4d datasets versus EK-55 and Gaze. All experiments are accomplished on NVIDIA A6000 GPUs.

### C.1 PREPROCESSING AND POSTPROCESSING

During preprocessing for fine-tuning, we empirically observe that using a single token to represent each verb or noun helps the performance of the fine-tuned model. Since the tokenization is handled by OpenAI's API, some verbs or nouns in the Ego4D's word taxonomy may span over multiple tokens. For example, the verbs "turn-on" and "turn-off" are two-token long, and the noun "coconut" is broken into three tokens. We attempt to minimize the token length of each Ego4D label without losing semantic information of the labels, which are important to leverage prior knowledge in the LLMs. As a result, we choose the first unique word to describe the verb and noun labels in the Ego4D label vocabulary. This preprocessing is performed for both the input action sequences, and the target action sequences for prediction. No additional task information is provided as input during fine-tuning, as we observe that the fine-tuning training examples themselves clearly specify the task.

We observe that LLM models demonstrate strong capability on following the instructions as specified by fine-tuning or in-context learning examples. However, the output sequences generated by the LLM are sometimes invalid, and post-processing is needed before the predictions can be evaluated by the edit distance metrics. Specifically, the standard Ego4D LTA evaluation protocol requires generating 5 sequences of long-term action predictions, each in the form of 20 verb and noun pairs.

We consider the following scenarios to trigger postprocessing on the LLM's predictions: (1) the output sequence length is different than expected (i.e. 20); (2) an action in the output sequence cannot be parsed into a pair of one verb and one noun; (3) a predicted verb or noun is outside the vocabulary. Table A7 shows the statistics on how often each type of incident would occur.

Instead of repeatedly sampling new responses from the LLM until it is valid, we choose to use a simple post-processing strategy: First, inspect each action in the predicted sequence, and remove the invalid ones. Second, truncate the sequence, or pad the sequence with the last predicted valid action until the sequence length is as desired. We then map the predicted words into the fixed noun and verb label space. This is achieved by retrieving the nearest neighbor of each word in our label vocabulary based on Levenshtein distance.

| Incident | % over all instances |
|---|---|
| Short Seq (1) | 2.74% |
| Long Seq (1) | 12.93% |
| Invalid Seq (2) | 10.44% |
| Invalid Verb (3) | 1.93% |
| Invalid Noun (3) | 2.62% |

Table A7: **Statistics on scenarios which require post-processing.** Invalid sequence refers to the scenario where the output sequence contains invalid strings that cannot be parsed into pairs of a verb and a noun. Invalid verb/noun refers to the scenario where the word is outside of the vocabulary. Both scenarios often imply wrong sequence length.

### C.2 BOTTOM-UP AND TOP-DOWN TRANSFORMER MODELS IMPLEMENTATION DETAILS.

For the bottom-up model that establishes our vision baseline, we use frozen CLIP Radford et al. (2021) image encoder (ViT-L/14@336px) as vision backbone. Each frame is encoded as a 768D

representation. For Ego4D, we use 3 layers, 8 heads of a vanilla Transformer Encoder with a hidden representation dimension of 2048. We use Nesterov Momentum SGD + CosineAnnealing Scheduler with learning rate 5e-4. We train the model for 30 epochs with the first 4 as warm-up epochs. All hyper-parameters are chosen by minimizing the loss on the validation set. In our top-down models, we used the compatible text encoder of the ViT-L/14@336px CLIP model as the text encoder. We used Adam optimizer with learning rate 2e-4.

For EK-55, we extracted features of all provided frames from the dataset. For the transformer, we use 1 layers, 8 heads with a hidden representation dimension of 2048. We use Adam optimizer with learning rate 5e-5. All hyper-parameters are chosen by minimizing the loss on the validation set. In our top-down models, we used the compatible text encoder of the ViT-L/14@336px CLIP model as the text encoder. We used Adam optimizer with learning rate 5e-5.

For Gaze, we use 1 layers, 8 heads of a vanilla Transformer Encoder with a hidden representation dimension of 2048. We use Nesterov Momentum SGD + CosineAnnealing Scheduler with learning rate 2e-2. All hyper-parameters are chosen by minimizing the loss on the validation set. In our top-down models, we used the compatible text encoder of the ViT-L/14@336px CLIP model as the text encoder. We used Nesterov Momentum SGD + CosineAnnealing with learning rate 15e-3.

## C.3 Recognition Model Implementation Details.

For the recognition model, we use frozen CLIP Radford et al. (2021) image encoder (ViT-L/14@336px) as vision backbone. For all the datasets, we randomly sample 4 frames from each segments. Each frame is encoded as a 768D representation. We use 3 layers, 8 heads of a vanilla Transformer Encoder with a hidden representation dimension of 2048. We use Nesterov Momentum SGD + CosineAnnealing Scheduler with learning rate 1e-3. We train the model for 40 epochs with 4 warm-up epochs. All hyper-parameters are chosen by minimizing the loss on the validation set.

## C.4 Inferring goal descriptors using LLMs

In Section 4.2, we describe an ablation study where goal descriptor is used for in-context learning. More specifically, we use GPT-3.5 Turbo and Llama2-chat-13B to infer the goal by giving recognized history actions and few samples as demonstrations with ICL as goal descriptors. An illustraion of the goals can be found in Figure A1. For the experiments in Table 1, the goal descriptors are added as LLM prompts in the form of `"Goal:<goal> Observed actions:<observed actions> => "`. We note that this ablation study poses an interesting contrast to the CoT experiments: Instead of asking the LLM to jointly infer the goal and predict the action sequences, the ablation experiment conducts goal inference and action prediction separately, and achieves slightly worse performance than CoT.

## C.5 Additional details on EPIC-Kitchens-55 and EGTEA Gaze + experiments

In the experiment section, we describe two setups of the LTA task on different datasets. For the Ego4D datasets, we measure the edit distance of future action sequence predictions while for EK-55 and Gaze, we measure the mean average precision for all actions to occur in the future. The distinction arises in how "action" is defined among these different datasets and setups. In Ego4D benchmarks, we follow the official definition of action, which a verb-prediction and a noun-prediction combining together to form an action. For EK-55 and Gaze, we adopt the task setup of Ego-Topo Nagarajan et al. (2020) in order to compare with previous works. In their setup they defined "action" as only the verb-prediction, excluding the noun-prediction in the final "action" prediction.

# D Additional Visualization and Analysis

## D.1 Examples of AntGPT Prediction

We provide four examples of bottom-up fine-tuned **AntGPT** in Figure A3, two of which are positive, and the other two negative. Correctly recognized actions from the video observations (outputs of the vision module, and inputs to the LLM) and future predictions (outputs of the LLM) are marked in green while wrong observations or predictions are marked red. We observe that **AntGPT** is able

to predict reasonable and correct future actions even when the recognized actions are not perfect. However, when most of the recognized actions are incorrect (Figure A3c), the predictions would also fail as expected. Finally, we observe that the bottom-up approach sometimes fail to implicitly infer the long-term goal based on the observed actions. In Figure A3d, the actor keeps on mixing the ice while the LLM predicts curry-cooking related activities.

## D.2 ADDITIONAL EXAMPLES OF ICL AND CoT

Similarly with Figure 2, Figure A4 shows two additional examples of bottom-up prediction with ICL and top-down prediction with CoT. We can further observe the influence of implicitly prediction future goal from the two examples. In the first example, due to the low quality of recognized actions from the video observation (repeating `play card`), the bottom-up prediction follows the observation pattern to repeat `play card` while in the prediction of top-down prediction using CoT to consider the goal, the model succeed to predict `take card` and `put card` alternatively which both matches the ground-truth future actions and common sense in the scene of card playing better, with the verb edit distance significantly goes down from 0.95 to 0.45. The second example shows a failure example of using CoT. In this video of "playing tablet", the top-down model predicts the goal as "using electronic devices", thus influencing the following future action prediction to hallucinate actions like `operate computer`. This behavior is intuitive, as an incorrect goal sampled by the LLM is likely to mislead the predicted actions that conditioned on it. We believe adding additional cues for goal inference, and explicitly model the different "modes" of the future to capture the goal diversity would mitigate this behavior.

## D.3 ADDITIONAL EXAMPLES OF COUNTERFACTUAL PREDICTION

In Figure A5, we demonstrate four additional counterfactual prediction examples. In order to predicting with explicit goal using ICL, we construct the prompts in the form of: `"Goal:<inferred/altered goal> Observed actions:<observed actions> => <future actions>"` We observe that switching the goals from the "inferred goal" to the "altered goal" indeed has huge impact on the predicted actions. This confirms that the goals are utilized by the LLM to predict (or plan) the future actions to take. This behavior is consistent with our observations in Figure A4.

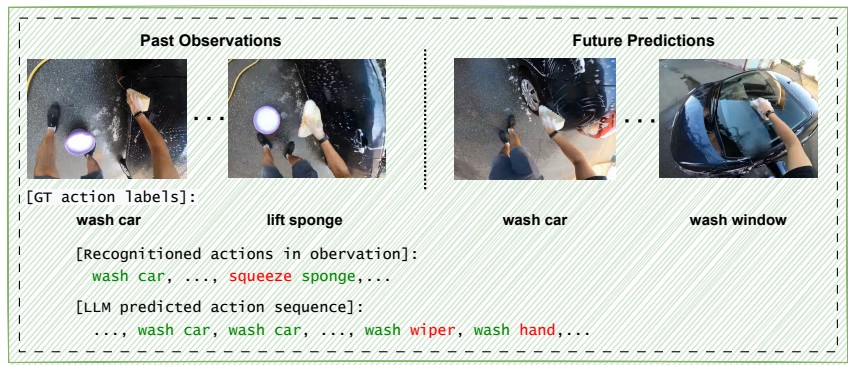

(a) Positive Sample 1

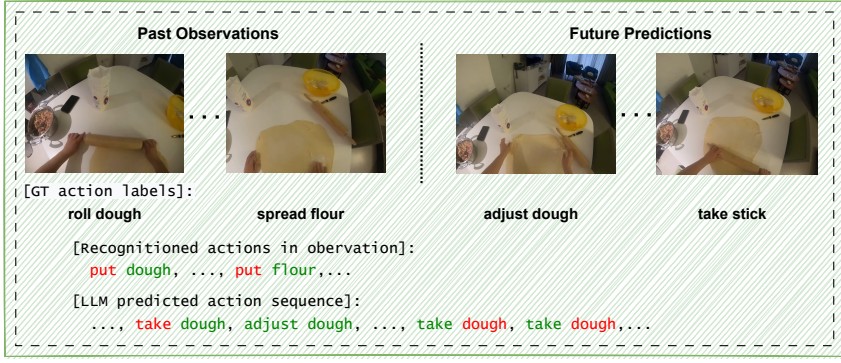

(b) Positive Sample 2

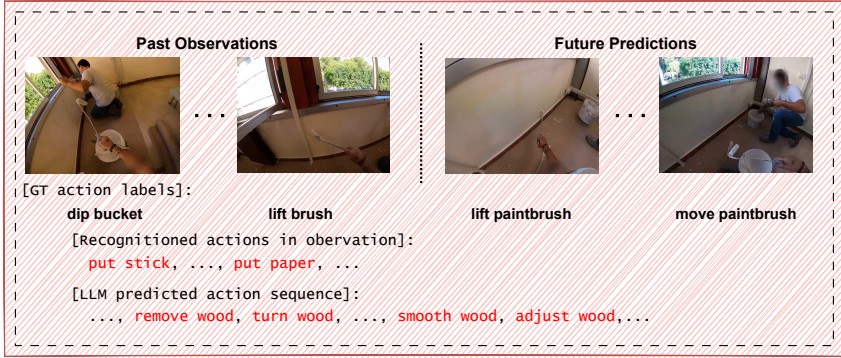

(c) Negative Sample 1

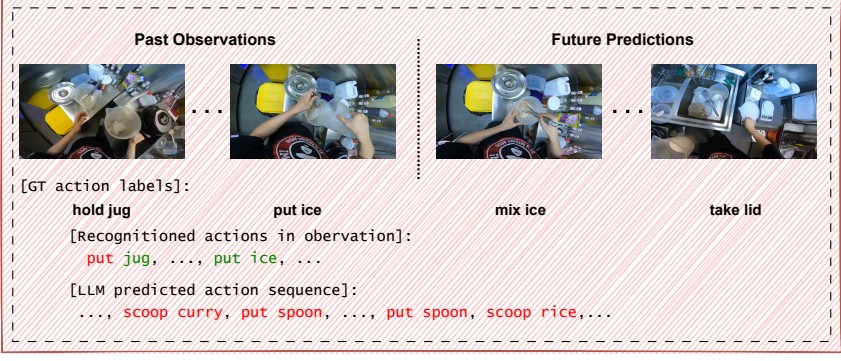

(d) Negative Sample 2

Figure A3: **Four examples of results from fine-tuned AntGPT.** The green word indicates correctly recognized actions (inputs to the LLM) and future predictions (outputs of the LLM), red indicates incorrectly recognized or predicted actions.

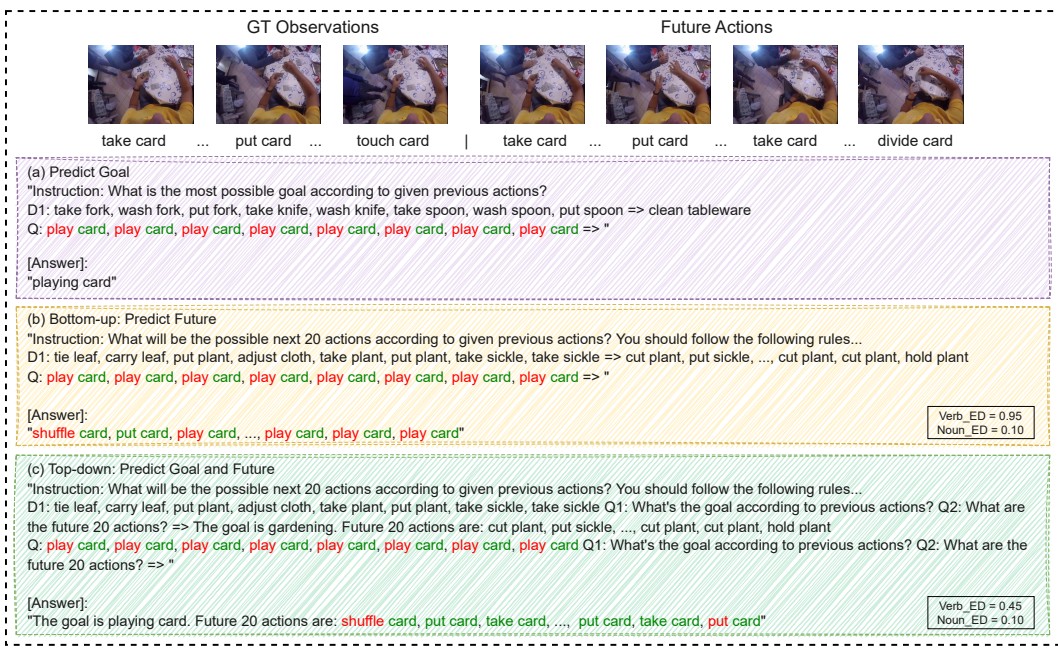

(a) Example 1, the ground-truth observations are [take card, put card, pack card, take card, put card, take card, put card, touch card], the ground-truth predictions are [take card, put card, take card, put card, pack card, take card, put card, pack card, take card, put card, take card, put card, put card, pack card, arrange card, take card, shuffle card, put card, take card, divide card].

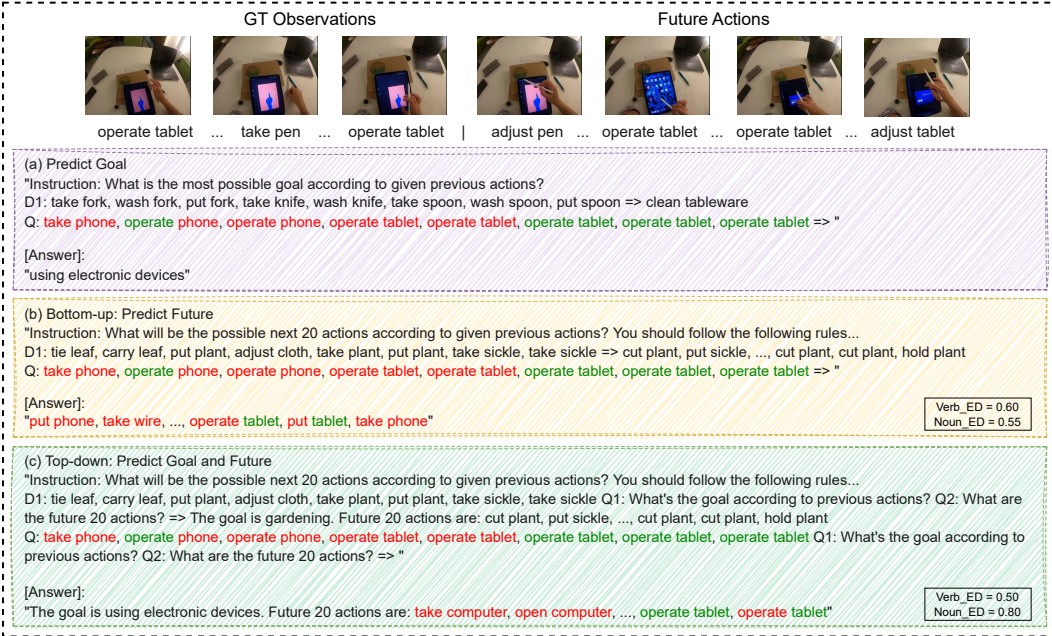

(b) Example 2, the ground-truth observations are [hold table, operate tablet, hold pen, touch pen, take pen, operate tablet, operate tablet, operate tablet], the ground-truth predictions are [adjust pen, hold pen, operate tablet, operate tablet, hold pen, operate tablet, operate tablet, operate tablet, operate tablet, operate tablet, move pen, operate tablet, lift tablet, put tablet, operate tablet, operate tablet, operate tablet, adjust tablet, operate tablet, adjust tablet].

Figure A4: **Promts and results of IcL and CoT**. Here we show three experiments: (1) Predict the final goal. (2) Botom-up predict future actions with ICL. (3) Top-down prediction by CoT prompting. Green word means correct observations or predictions and red means wrong.

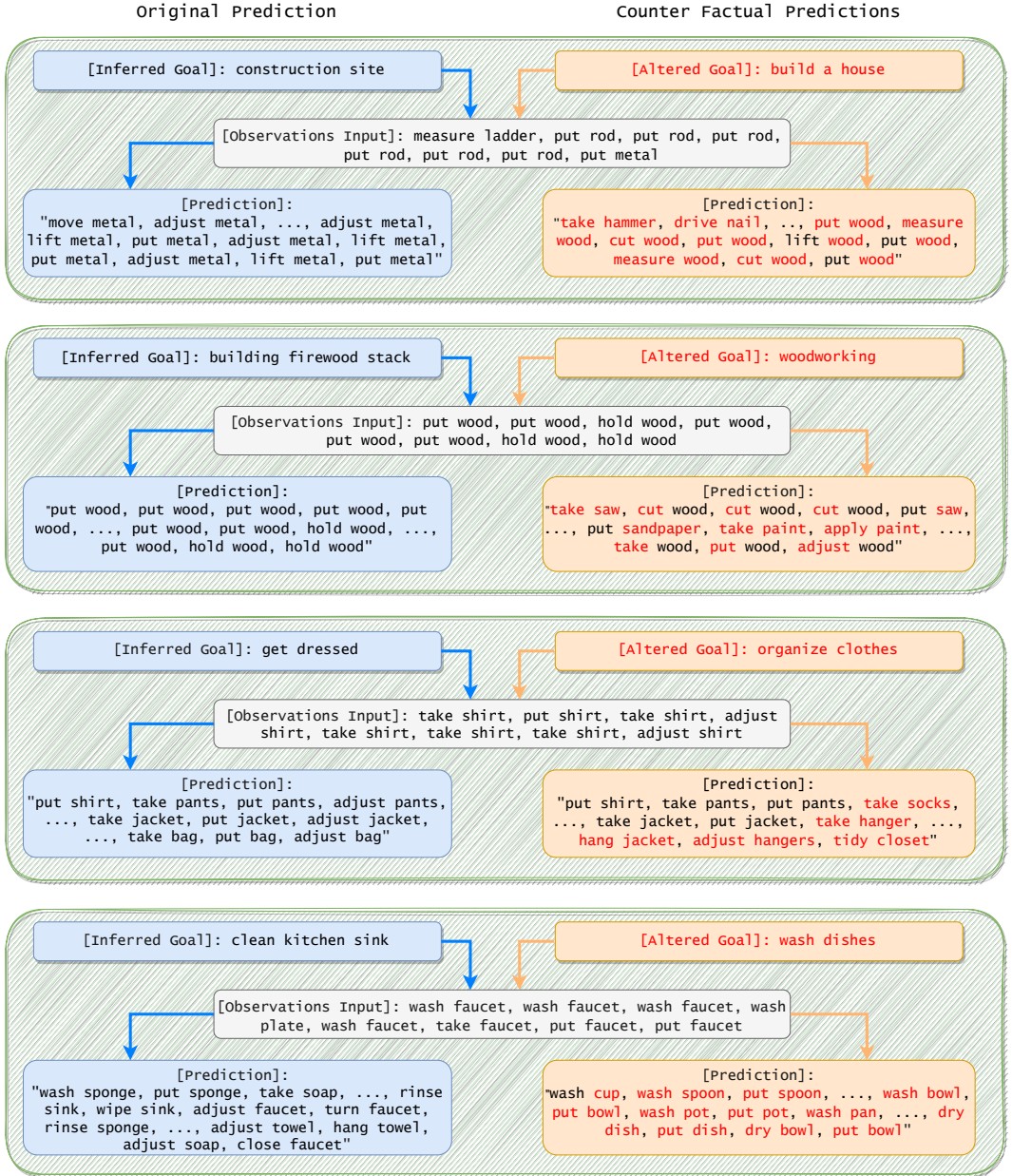

Figure A5: **Four examples of counterfactual prediction**. LLM predict based on same observations but different goals. On the left shows the results with the "inferred goal" generated by GPT-3.5-Turbo according to observations generated by recognition model while the right part shows results with manually designed "alternative goal".

