# OpenReview forum: "AntGPT: Can Large Language Models Help Long-term Action Anticipation from Videos?"
_ICLR.cc/2024/Conference — ICLR 2024 poster_

### Official Review · Reviewer_R7fa · 2023-10-27

**Soundness:** 3 good
**Presentation:** 3 good
**Contribution:** 2 fair
**Rating:** 5
**Confidence:** 4

**Summary:**

This paper proposes a method named AntGPT to investigate if large language models encode useful prior knowledge on bottom-up and top-down long-term action anticipation. Experimental results on Ego4D LTA, EPIC-Kitchens-55, and EGTEA GAZE+ benchmarks indicate the effectiveness of the proposed method.

**Strengths:**

-	The motivation is clear, and the usage of large pre-trained models is interesting.
-	Experimental results on multiple benchmarks indicate the effectiveness of the proposed method.

**Weaknesses:**

-  The novelty is limited, AntGPT is a straightforward application of large language models to the action anticipation problem.
- In Table 1, the improvement of the proposed method is marginal.
- The proposed method relies on the classification results of previous fragments, and classification errors may lead to the accumulation of errors and affect the prediction results.

**Questions:**

Please refer to the Weaknesses for more details.

---

> ### Author Response · Authors · 2023-11-22
>
> We appreciate your detailed and constructive feedback!
> >***Q1: The novelty is limited***
>
> We respectfully disagree. We view the simplicity of our design and the effectiveness of our approach as desirable properties and contributions. In order to arrive at our simple and effective design, three fundamental research questions need to be addressed:
> + Does goal conditioning benefit LTA? If so, can we utilize LLM to infer the goals without supervision?
> + Do LLMs capture prior knowledge to model temporal dynamics?
> + Can we condense LLM’s prior knowledge and temporal modeling ability to a compact neural network for efficient inference?
>
> Our paper strives to answer these questions and design corresponding experiments accordingly. Despite its simplicity, to the best of our knowledge we are the first to demonstrate the effectiveness of the action-based video representation for long-term action anticipation, and how to properly leverage and condense the prior knowledge encoded by LLMs for the task.
>
> >***Q2: The improvement of table 1 is marginal***
>
> We would like to highlight that Table 1 only studies one of our contributions, namely goal-conditioned LTA, where the goals are inferred by LLM via few-shot in-context learning. We show that the goals inferred by LLMs (Llama-2 and GPT-3.5) lead to consistent improvements across all three benchmarks, especially for the rare actions in EK-55 and EGTEA. We also show that the models based on inferred goals even approach the performance when oracle goals are used.
>
> When the whole system is evaluated and compared against the previous state-of-the-art methods, we show in Table 6 and 7 that AntGPT outperforms all previous methods by large margin. We also report the variances of AntGPT’s performance to further reinforce that the gains are statistically significant.
>
> >***Q3: Rely on classification results***
>
> The reviewer is correct that as expected, action recognition accuracy would have an impact on the LTA performance. We recognized this and performed an ablation study when oracle (ground truth) actions instead of recognized actions are used by the LLM. This oracle model achieves 0.679 verb ED and 0.617 noun ED on the validation set of Ego4D v1. This result shows that: (1) The LTA performance is indeed reliant on the action recognition accuracy; (2) AntGPT with recognized actions achieves competitive performance when compared to the oracle actions; (3) The LTA task remains challenging even when oracle actions are used, we attribute this to the challenge of modeling temporal dynamics.
>
> In addition, Table A2 (Appendix) studies on the impact of different visual backbones (hence action recognition performance). We observe that as expected, better recognition models do benefit future action prediction but this influence is moderate. Besides, the final results also show that our proposed method achieves strong performance on all 3 benchmarks even with imperfect recognition results.

---

### Official Review · Reviewer_jEKR · 2023-10-28

**Soundness:** 3 good
**Presentation:** 3 good
**Contribution:** 3 good
**Rating:** 8
**Confidence:** 4

**Summary:**

The paper studies whether and how large language models can assist long-term action anticipation (LTA) from videos. To do so, the paper introduces AntGPT, an approach to perform LTA based on a mixture of a bottom up (direct prediction from observations) and top down (goal prediction, then action sequence prediction conditioned on goals) approaches. The paper also shows that the large language models used in the approach can be effectively distilled to smaller models for convenience. Extensive evaluations assess the influence of each component on final performance. The proposed approach is shown to outperform state-of-the-art approaches on major datasets.

**Strengths:**

The question, whether LLMs can assist action anticipation, is surely interesting.

I found the paper informative and the study well designed to answer the research questions stated in the introduction.

Experiments are extensive and designed to be informative for the reader, beyond showcasing the abilities of the proposed approach.

The final model is shown to outperform previous approaches on the main datasets.

I appreciated the "limitations" section.

**Weaknesses:**

While I believe the paper is sound and an interesting contribution, one point is not clear to me:

In-context (few-shot) learning is used to anticipate future actions with LLMs. The idea of using few-shot seems to be motivated by constraints due to LLMs rather than due to the task. Indeed, while one may use the entire training set as a set of examples, this would lead to a large prompt, which could be both impractical and detrimental to performance. However, in principle, it would make perfect sense to avoid few-shot learning altogether and resort to some form of supervised learning. While I understand the constraints, I feel this part is not well discussed in the paper. Also, how are few-shot examples selected in practice?

Minor comments:
- the "visual procedure planning" paragraph in the related works section is not very easy to follow. It seems like the paragraph states what visual procedure planning is related to, but it does not give a clear definition of what it is intended by the term "visual procedure planning". I'd suggest revising.
- in the same section, the reference (Mascaro et al.) is missing the publication year
- "an few-shot high-level goal predictor" -> "a few shot ..."
- at page 5, the paper states "we either use the video metadata as pseudo goals when it is available, or design the goals manually". It is not clear to me what "design the goals manually" means, and this does not seem to be well described in the main paper.
- the paper relies on CLIP, but it would have been interesting to see how the use of other ego-specific video-language models such as EgoNLP, LaViLa or EgoNLPv2 would perform in this context
- while experiments are performed on EPIC-KITCHENS-55, no experiments are performed on EPIC-KITCHENS-100. I believe experiments on EK55 are relevant enough, but EK100 is a larger dataset (of which EK55 is a subset) and a much denser and cleaner one, so it would seem a perfect test bed for LTA. I'd encourage the authors to consider it for future experiments.
- In Table 1, there is a star (*) next to "oracle goals", but I could not find the explanation of this symbol anywhere
- In table 5, "from-scratch" is reported for the second model as setting. Does this mean that the model was randomly initialized? From the text it seems that it was still pre-trained on some text corpus. Can the authors clarify (or fix) this?

**Questions:**

The authors could discuss on few-shot learning as referenced above and comment on the most relevant minor comments.

---

> ### Author Response · Authors · 2023-11-22
>
> Thank you for your constructive feedback! We address your concerns in detail below and revise our paper accordingly:
>
> >***Q1: Few-shot learning***
>
> We would like to clarify the purpose of our few-shot learning experiments. Since we use LLMs as the temporal model, the experiment aims to understand if in-context learning is also more desirable than direct gradient updates (as have been shown for language-based tasks with LLMs), when one only has a few examples to “supervise” the LTA task.
>
> The few-shot learning scenario arises when one has access to a reasonable action recognition model (e.g. those trained on Kinetics, an action recognition dataset of temporally trimmed 10-second video clips) but only has a limited amount of examples to demonstrate the temporal dynamics. We acknowledge that the Ego4D dataset is not an ideal benchmark for this scenario, since the actions are already densely annotated over time, and can be used to provide “full” supervision for both action recognition and anticipation. Nonetheless, we believe that we can realistically simulate the few-shot LTA scenario by **randomly** select the LTA examples (i.e. simulate the scenario where one only has access to those few examples), in order to understand the desirable approach to perform few-shot LTA with LLMs. And that is indeed the setup we adopt in the experiments.
>
> >***Q2: Minor comments***
>
> We would like to express our appreciation for these comments.
>
> + For the “manually designed goals”, we describe how we design the goals manually in our response to Reviewer Utid Q3 and update the description in our revision.
>
> + For ego-specific video-language models , we compare with HierVL, which is a recent sota ego-specific video-language model (based on EgoVLP). Besides, we also conducted an ablation study to compare CLIP and EgoVLP as backbone vision models in Table A2 in the supplementary, the results show that incorporating egocentric visual features indeeds brings some improvement.
>
> + For experiments on EK100, thank you for the suggestion! We’ll consider EK-100 as another major benchmark for future work, we picked EK-55 for the ease of comparisons with previously published results.
>
> + In table 5 the “from-scratch” means the 91M model is initialized randomly. In fact in this paper the 91M model is always trained from scratch with random initialization. We’ll clarify it in the revised version.
>
> + We have made the corresponding changes in our revision to address the other points raised by the reviewer.

---

> > ### Comment · Reviewer_jEKR · 2023-11-22
> >
> > I'd like to thank the authors for clarifying the point on in-context learning. I now understand the main rationale, but I think this is not very well described in the current version of the paper and I'd appreciate if the authors would improve this part of the paper in the revised version.
> >
> > While subsampling Ego4D makes sense as a testbed, a more appropriate setup would be to train the recognition model on dataset A, then test it on dataset B, for which only few action instances are annotated. I understand that this brings other domain adaptation challenges, but it would probably bring a more convincing argument.

---

> > > ### Author Response · Authors · 2023-11-22
> > >
> > > We would like to thank the reviewer for your constructive feedback and prompt reply!
> > >
> > > As suggested, we have updated the revision to clarify the motivation and setup of our few-shot learning experiments (see page 16). We fully agree with the setup you suggested is interesting and more convincing for few-shot learning. We are working on utilizing zero-shot action recognition models (e.g. VideoCLIP, Xu et al., EMNLP 2021) for the few-shot LTA experiments. Due to the time constraint, we will report them in the final version.

---

### Official Review · Reviewer_v3Zh · 2023-10-29

**Soundness:** 3 good
**Presentation:** 3 good
**Contribution:** 2 fair
**Rating:** 6
**Confidence:** 4

**Summary:**

This paper investigates using large language models to help action anticipation in bottom-up or top-down manners. For the bottom-up approach, future actions are predicted using observed videos or actions. For the top-down approach, the prediction of future actions is additionally conditioned on the goal. The LLMs are used by prompting or fine-tuning.

**Strengths:**

1. The formulation of decomposing to bottom-up and top-down approaches is interesting and intuitive.
2. Performance gains are observed on several datasets and ablation studies are good in general.
3. The writing and presentation is good in general.

**Weaknesses:**

1. The interpretation of the results should be cautious. The performance gain is evident but not surprising since additional labels of goals and additional knowledge from LLMs are introduced. Although the boundary of using additional information has become vague since the popularity of LLMs, I still believe we should be cautious when comparing a new method with LLMs and previous methods without LLMs.

2. Using LLMs for action anticipation makes me raises the question about the fundamental positioning of this task. In this sense, the action anticipation is mainly a language-understanding task rather than a vision task. If we look at Table 2, it is better to use actions as inputs than using visual features. If we still treat action anticipation as a vision task, what is the role and the importance of the visual modality? If we treat it as a language task, the contribution of this paper will be limited since it mainly utilizes prompting and fintuning of LLMs. If we treat it as a multi-modal task, we need to see more insights and experiments about the interaction between modalities.

**Questions:**

I don't have too many questions about the technical part of this paper. I would appreciate comments on how should we position the task of action anticipation in the following research. Should this task actually be a compound task of action recognition (for observed video) and language understanding for anticipation? Or should we still study this task as a standalone task?

---

> ### Author Response · Authors · 2023-11-22
>
> Thanks for your constructive feedback and inspiring questions!
>
> > ***Q1: The performance gain is evident but not surprising since additional labels of goals and additional knowledge from LLMs are introduced.***
>
> We would like to clarify the significance of the performance gains achieved by AntGPT as follows:
>
> + We consider our proposed goal conditioned framework (top-down) as a novel perspective for the LTA task and a contribution on itself. Prior work either adopts the bottom-up approach or uses latent goals. Instead, our Table 1 demonstrates the effectiveness of explicit goal-conditioning, where the goals are inferred by LLMs through in-context learning. Although additional goal annotations are used, it is worth highlighting that we only use few-shot examples for the goal generation: Only 12 examples are used for Ego4D (over 3,000 hours), EK-55 (55 hours), and EGTEA (26 hours), respectively.
>
> + Although it might appear to be intuitive for some that LLMs might be helpful for the LTA tasks, we believe our work strives to understand if and how they benefit for the LTA tasks, via extensive ablation studies (Table 1 for goal generation, Table 2 and 3 for temporal modeling, and Table 4 for input representations). These studies allow us to propose a compact model (91M) that is able to outperform its LLM counterpart (91B). Finally, we did compare with two LLM-based approaches in Table 6, where AntGPT outperforms both approaches.
>
> >***Q2: The role and importance of visual modality. Fundamental positioning of the LTA task.***
>
> We thank the reviewer for the excellent and thought-provoking questions! We view demonstrating the effectiveness of action-based representations for modeling temporal dynamics of human activities as one of the major contributions of AntGPT. This provides a natural interface for video understanding tasks to benefit from prior knowledge (e.g. through goals, or few-shot in-context learning) encoded by LLMs. Nonetheless, we view AntGPT as a **multimodal** framework that first leverages the visual inputs to generate interpretable and discrete action tokens, which are then processed by LLMs to model temporal dynamics. In this multimodal framework, text serves as a representational bottleneck between perception and temporal “reasoning”.
>
> AntGPT can be naturally generalized to relax the representational bottleneck, for example by adding visual embeddings alongside the action labels as inputs to the temporal model. We conduct such an analysis by concatenating CLIP embeddings to the discrete action labels, and add a linear projection layer to map the visual embeddings to the language input space, a standard practice adopted by BLIP-2 (Li et al.), AnyMAL (Moon et al.), and other work. The results are shown in the table below.
>
> | Input                  | Verb      | Noun      | Action    |
> |------------------------|-----------|-----------|-----------|
> | Action (AntGPT)        | 0.661     | 0.651     | 0.878     |
> | Action + CLIP features | **0.643** | **0.650** | **0.868** |
>
> After combining visual features and discrete action labels, we observe a moderate level of performance improvement compared with the original AntGPT results (from Table A4). This shows that: (1) our proposed framework can indeed be naturally extended with a hybrid “bottleneck” that includes both text and visual inputs; (2) for the LTA benchmark, we observe the gain of directly incorporating vision features is moderate, which confirms the strength of our proposed original AntGPT with discrete action representation.
>
> In summary, we believe action anticipation should still be studied as a standalone task. Despite AntGPT’s strong performance across multiple benchmarks, we believe better and more detailed visual representation (e.g. gaze and pose of the actors) are essential for solving the task, and AntGPT can be naturally extended to achieve this goal as demonstrated above. Other possible improvements include incorporating fine-grained, object-centric visual representations, and even to directly backpropagate the gradient to the visual encoders.

---

> ### Comment · Reviewer_v3Zh · 2023-11-22
> **Thanks**
>
> Thanks for the rebuttal. It partially answers my questions.
>
> It is helpful to see some early investigation on the multi-modality issue. I would like to slightly raise my score.
>
> But to have a clear acceptance, we need a more principled multimodal approach or rethink the task definition.

---

> > ### Author Response · Authors · 2023-11-23
> > **Thank you**
> >
> > Dear reviewer v3Zh,
> >
> > Thank you so much for your prompt reply to our rebuttal, and we appreciate your score increase!
> > We believe AntGPT is a principled multimodal approach that demonstrates the effectiveness of goal-conditioned action anticipation, modeling temporal dynamics with action-based representation, and how LLMs may assist both. We acknowledge and agree with the reviewer that further explorations on more directly incorporating visual information into the representational bottleneck would make interesting and important future work. Thanks!
> >
> > Best,
> > The authors

---

### Official Review · Reviewer_Utid · 2023-10-31

**Soundness:** 3 good
**Presentation:** 1 poor
**Contribution:** 3 good
**Rating:** 6
**Confidence:** 3

**Summary:**

This paper proposes to utilize large language models for long-term action anticipation. CLIP visual features are extracted from each video segment and a sequence of action labels are predicted, which are input to the LLM. In-context learning is used to predict the final goal, which is embedded with the CLIP text encoder. Experiments on three datasets show the efficacy of the proposed method. The authors also show the distilled smaller models can be effective as well.

**Strengths:**

1. The idea of using LLMs for long-term action anticipation is interesting and the intuition makes sense.
2. Experiments are thorough. The authors compare multiple usages of LLMs for the LTA task, including in-context learning and fine-tuning.
3. The proposed method achieves SOTA on three benchmarks.

**Weaknesses:**

1. The paper writing/organization can be improved. The method description part of AntGPT is quite confusing. For example, an overview of how the separate parts of Figure 1 (b-e) are connected to each other will help. Also, for the experiment section, putting “comparison with state-of-the-art” to the front might be better IMO.
Other minor presentation issues: Some text in Figure 1 is too small to read.
2. There is no error analysis. When does the proposed method fail and why?

**Questions:**

1. The goals in Figure 2 are sometimes simply a repetition of the last action labels. Can the authors elaborate on how the goals are defined (manually annotated)? Should they be the last action of the videos?
2. In Table 5, the distilled model is even better than the original model. How is that possible? Can the authors provide more explanation on that?

------------Post rebuttal comments: the authors have addressed my questions and concerns. I have increased my score.

---

> ### Author Response · Authors · 2023-11-22
>
> We thank reviewer Utid for your helpful comments and constructive feedback! Due to character limits, we'll answer your question with two comment pages.
> > **Q1: The paper writing/organization can be improved.**
>
> Thank you for the feedback! We have updated Figure 1 as suggested in revision, to emphasize the connections among different panels and made the fonts larger. Notably, Figure 1(a) provides an overview of the AntGPT framework, which illustrates how the action recognition model (blue), temporal model (orange), and goal generator (red) are used in the bottom-up, and top-down LTA frameworks, respectively. The color coding is reused for Figure 1(b-d) to indicate their connections. Figure 1(b) provides a zoomed-in view of the goal generator to illustrate how “goals” are defined in the context of our work. Figure 1(c) discusses a special case of Figure 1(a), when in-context examples (as opposed to explicitly computing gradient updates) are provided to the temporal model for few-shot LTA. Figure 1(d) discusses another use case for knowledge distillation.
>
> We acknowledge your suggestion on the state-of-the-art results, our intention was to first provide answers to our core research questions, and context information on how AntGPT achieves state-of-the-art performance.
>
> We hope our edits address both of your feedback on the paper’s presentation, and we look forward to your additional feedback!
>
> > **Q2: Lack of error analysis**
>
> We provided error analysis in the appendix due to space limitations. Specifically, Figure A1 illustrates how the goal-conditioning would improve long-term action anticipation performance. Figure A4 presents two positive and negative examples when the LLM for temporal dynamics modeling is fine-tuned. Figure A5 presents examples on how chain-of-thought prompting may help (a) or hurt (b) LTA performance. Finally, Figure A6 (Section B1) presents the counterfactual examples.
>
> Through these qualitative analysis, we can see that long-term action anticipation is a very challenging task, and nonetheless AntGPT is capable of predicting correct future actions, or making reasonable mistakes, even when the recognized actions are not perfect. However, we do observe that when most of the recognized actions are incorrect (Figure A4c), the predictions would also fail as expected. Other failure modes include the hallucination effect of LLMs (Figure A4d). Figure A5b shows a failure example of chain-of-thoughts prompting. In this video of “playing tablet”, the top-down model predicts the goal as “using electronic devices”. As expected, an incorrect goal is likely to mislead the predicted actions conditioned on the goal. We believe adding additional supervision (when available) to the goal generator, and improving the LLM backbone to reduce hallucination, would both help mitigate the errors.

---

> ### Author Response · Authors · 2023-11-22
>
> (continues)
> > **Q3: How are the goals defined?**
>
> A goal is defined as the intent / purpose of a human actor’s behaviors. In the context of daily household activities, a goal often corresponds to the task the actor is trying to accomplish (e.g. take the action “mix eggs” to accomplish the goal of “making fried rice”). The desired goals are hence not supposed to be always a simple repetition of the last observed action, but to provide a description of the task. To meet our definition of desired goals, we collect the few-shot examples for goal generation with in-context learning as follows: For Ego4D, we manually annotate goals as the task descriptions by watching the videos. For EGTEA, we use the video title (e.g. “cook cheeseburger”). For EK-55, we use the video descriptions from the dataset (e.g. “cook instant noodles for lunch”). The newly added **Figure A3** in appendix shows goals we used to construct the in-context examples for goal generation.
>
> Figure 2 shows the predicted goals based on the in-context examples. In Figure 2a, the last observed action “paint (with) paintbrush” is to accomplish the task “paint the door”; in Figure 2b, the last observed action “clean mat” is to accomplish the task “clean car and mat”. Although the inferred goals appear to correlate to the future actions, we still view them as positive examples since they accurately describe the high-level tasks nonetheless.
>
> > **Q4: Distillation results are better than LLMs.**
>
> The student model is optimized by both distillation loss and language modeling loss as shown in Figure 1d. We conducted comparing experiments showing that distilling results with only distillation loss without loss from ground-truth future actions are strictly worse than the teacher model’s performance (shown in the table listed below). We thus attribute the better performance of the student model to: (1) the integration of knowledge from the teacher model through distillation; (2) the optimization of a student model with task-specific ground-truth labels (in comparison, the LLMs are pre-trained as “generalists”, then adapted to the LTA tasks).
>
> | Loss      | verb       | noun       | action     |
> |-----------|------------|------------|------------|
> | dist      | 0.7044     | 0.6788     | 0.9041     |
> | dist + lm | **0.6649** | **0.6752** | **0.8826** |

---

> ### Comment · Reviewer_Utid · 2023-11-23
>
> Thanks for the reply. The authors have addressed all my questions and concerns.

---

> > ### Author Response · Authors · 2023-11-23
> > **Thank you**
> >
> > Dear reviewer Utid,
> >
> > Thank you so much for your prompt reply to our rebuttal, and your confirmation that our rebuttal has addressed all your questions and concerns.
> >
> > Would you please let us know if our submission, considering our rebuttal and revision, would meet your criteria for a higher score for presentation, and the overall rating? Thank you so much for your consideration!
> >
> > Best,
> > The authors

---

### Meta-Review · Area_Chair_bMPL · 2023-12-11

**Metareview:**

This paper received somewhat mixed but overall positive ratings (8, 6, 6, 5). The reviewers liked that the paper studies a timely and interesting problem and proposes an intuitive LLM-based solution for action anticipation. They also acknowledged that the experiments are thorough and convincing. Initial concerns are well addressed in the rebuttal; as a result, two reviewers increased their ratings from 5 to 6. Overall, the paper shows an interesting use case of LLMs for action anticipation in videos, and the idea is convincingly demonstrated with empirical results.

**Justification For Why Not Higher Score:**

Utilizing pretrained LLMs for addressing vision tasks is gaining widespread popularity. This paper presents a well-thought-out pipeline for action anticipation. However, from a high-level perspective, it could be viewed as an incremental contribution to the existing body of literature.

**Justification For Why Not Lower Score:**

The merits, as highlighted by the reviewers, make it a great addition to the proceedings.

---

### Decision · Program_Chairs · 2024-01-16

Accept (poster)